# Design and Development of a Non-Contact ECG-Based Human Emotion Recognition System Using SVM and RF Classifiers

**DOI:** 10.3390/diagnostics13122097

**Published:** 2023-06-16

**Authors:** Aftab Alam, Shabana Urooj, Abdul Quaiyum Ansari

**Affiliations:** 1Department of Electrical Engineering, Jamia Millia Islamia, Delhi 110025, India; aqansari@jmi.ac.in; 2Department of Electrical Engineering, College of Engineering, Princess Nourah bint Abdulrahman University, P.O. Box 84428, Riyadh 11671, Saudi Arabia

**Keywords:** electrocardiogram (ECG), emotion classifier, emotion recognition system, human-machine interaction (HMI), SVM, RF

## Abstract

Emotion recognition becomes an important aspect in the development of human-machine interaction (HMI) systems. Positive emotions impact our lives positively, whereas negative emotions may cause a reduction in productivity. Emotionally intelligent systems such as chatbots and artificially intelligent assistant modules help make our daily life routines effortless. Moreover, a system which is capable of assessing the human emotional state would be very helpful to assess the mental state of a person. Hence, preventive care could be offered before it becomes a mental illness or slides into a state of depression. Researchers have always been curious to find out if a machine could assess human emotions precisely. In this work, a unimodal emotion classifier system in which one of the physiological signals, an electrocardiogram (ECG) signal, has been used is proposed to classify human emotions. The ECG signal was acquired using a capacitive sensor-based non-contact ECG belt system. The machine-learning-based classifiers developed in this work are SVM and random forest with 10-fold cross-validation on three different sets of ECG data acquired for 45 subjects (15 subjects in each age group). The minimum classification accuracies achieved with SVM and RF emotion classifier models are 86.6% and 98.2%, respectively.

## 1. Introduction

Emotion is an important aspect of human consciousness, which drives our mental state even subconsciously. The emotional state ensures mental well-being, as well as our overall health. The human emotional state is a result of chemical changes in the brain which affect the whole body and its overall expression and actions. They affect feelings, which certainly act as an important parameter which differentiates humans from other species. We feel a diverse range of emotions which may be often situational and might be triggered by outside events. A constant state of sadness for a long time causes depression. A state of severe mental illness may also result in physical illness. When a person is angry, the temperature of the body increases and may even lead to shivering. The blood pressure of a person fluctuates in cases of intense happiness and sadness. A state of intense fear results in sweating and an increase in heart rate. The state of disgust and surprise may lead to a reduction in the heart rate. A pleasant activity relaxes our mood and reduces stress which leads to a reduction in heart rate in comparison to a hyper state. The rhythm of the heart changes with emotions. Researchers have tried to correlate various facial features, speech signals, and audiovisual features, as well as physiological signals such as EEG (electroencephalogram), ECG, GSR (galvanic skin response), and respiration signals, with changes in emotions. Broadly, human emotion recognition systems are categorized into non-physiological- and physiological-based systems. Non-physiological systems utilize facial expressions, speech, audio, and video of the subject when exposed to elicit emotions through an external stimulus. As these features can be masked, for example, a happy person can pretend to have a serious and sad facial expression, as well as a sad person can pretend to have a smiling face, a physiological-based system is also of merit. This second method utilizes physiological signals such as ECG, EEG, GSR, and breathing signals as feature datasets to classify human emotions. The physiological signals are involuntary in their source of generation, hence they cannot be masked or controlled by the subject. Much work has been reported using non-physiological methods of emotion recognition. Our motivation in this work was to utilize the minor changes in ECG waveform caused by changes in emotional state to classify seven different kinds of emotions. An ECG is a quasi-periodic bioelectrical signal which is an electrical activity of the heart acquired in the form of skin potential. This work reports the design and development of a unimodal (ECG)-based contactless human emotion recognition system. The functional block diagram of the proposed system is shown in Figure 1.

## 2. Related Work

With the advancement of technology, healthcare services have become more patient-oriented. The implementation of IoT (Internet of Things) and AI (artificial intelligence) with ML (machine learning)-based systems enables us to provide the required preventive care. They are widely used to develop smart healthcare systems for biomedical applications. The detection of diseases, from the very dangerous to the least dangerous, is being conducted using ML techniques. Machine-learning models can process a huge amount of patient data, including medical history, from many hospitals very quickly and are used in the detection and classification of diseases. For example, H. Zhu et al. presented an effective Siamese-oriented region proposal network (Siamese-ORPN) for visual tracking, and the authors proposed an efficient method of feature extraction and feature fusion [1]. W. Nie et al. illustrated a dialogue emotion detection system based on the variation of utterances represented in the form of graph models. The general knowledge conversation gestures in addition to the dialogue were utilized to enhance emotion detection accuracy. A self-supervised learning method was used and optimization techniques were proposed [2]. Zhiyong Xiong et al. developed a physiotherapy tool named SandplayAR and evaluated its impact on 20 participants with mild anxiety disorder. They showed the potential of the SandplayAR system with augmented sensory stimulation to be used as an efficient psychotherapy tool [3]. Rehab A. Rayan et al. briefly reviewed the potential of IoT and AI with ML technologies in biomedical applications. It was established by the authors that technology accelerates the transition from hospital-centered care to patient-centered care. With the development of ML techniques, healthcare devices are capable of handling, storing, and analyzing big data automatically and quickly [4]. Giacomo Peruzzi et al. developed a small, portable microcontroller-based sleep bruxism detection system. The system is capable of classifying the sound of bruxism from other audio signals and can detect the condition remotely at the patient’s house using a CNN-based ML technique [5]. Yuka Kutsumi et al. collected bowel sound (BS) data from 100 participants using a smartphone. They developed a CNN model which is capable of classifying BSs with an accuracy of 98% to comment on the gut health of a person [6]. Renisha Redij et al. also illustrated the application of AI in classifying BSs. The literature explains the relevance and potential of AI-enabled systems to change the GI practice with patient care [7]. Rita Zgheib et al. reviewed the importance of artificial intelligence with machine learning and semantic reasoning in the current scenario of digital healthcare systems. They illustrated and analyzed the relevance of AI and ML technologies in handling the COVID-19 pandemic [8]. Yash Jain et al. developed an ML-based healthcare management system which can act as a virtual doctor to provide a preliminary diagnosis based on the information provided by the subject. The CNN-based ML technique was utilized as well as a GUI interface developed. The system also includes an emotion classifier system. Such technologies are surely going to contribute to digital healthcare management and systems in the future [9].

### 2.1. Non-Physiological Signal-Based Emotion Classifiers

The non-physiological method of emotion classification involves inputs as responses such as speech, audio, video, and facial expressions corresponding to various emotions. K. P. Seng et al. reported an audiovisual feature-extraction algorithm and emotion classification technique. An RFA neural classifier was used to fuse kernel and Laplacian matrices of the visual path. The emotion recognition for seven types of expressions achieved an accuracy of 96.11% on the CK+ database and 86.67% on the ENTERFACE05 database [10]. Facial expressions can be masked by the subject who may control his/her reactions. T. M. Wani et al. reviewed various speech emotion recognition (SER) techniques in brief. The publicly available databases of speech signals in different languages include the list of models developed. Various feature extraction algorithms are explained and illustrated. The relevance and details of classifiers, such as GMM, HMM, ANN, SVM, KNN, DNN, etc., in speech-based emotion recognition have been illustrated [11]. The research gap includes the selection of robust features and machine-learning-based classification techniques to improve the accuracy of emotion recognition systems. M. S. Hossain et al. illustrated a real-time mobile-based emotion recognition system with fewer computational requirements. Facial video was used as data which were acquired using the inbuilt camera of a mobile phone and bandlet transform was realized with the Kruskal–Wallis feature selection method. The CK and JAFFE database were used to achieve a maximum accuracy of more than 99% [12]. Mobile systems have limitations in their data handling capacity and computational efficiency. S. Hamsa et al. utilized a speech correlogram and used an RF-classifier-based deep-learning technique to recognize human emotions in a noisy and stressful environment. English and Arabic datasets were used and processed to extract features after noise reduction. The four different datasets used were the ESD private Arabic dataset and the SUSAS, RAVDESS, and SAVEE public English datasets, and an average accuracy of more than 80% was achieved [13]. S. Hamsa et al. proposed an emotionally intelligent system to identify the emotion of an unknown speaker using energy, time, and spectral features for three distinct speech datasets of two different languages. Random forest classifiers were used to classify six different kinds of emotions and achieved a maximum accuracy of 89.60% [14]. L. Chen et al. proposed a dynamic emotion recognition system based on facial key features using an Adaboost-KNN adaptive feature optimization technique for human–robot interaction. Adaboost, KNN, and SVM were used for emotion classification. They reported a maximum accuracy of 94.28% [15]. S. Thuseethan et al. proposed a deep-learning-based unknown facial expression recognition technique. They presented a CNN-based architecture for efficient testing results. Model efficacy was evaluated using the benchmark emotion dataset and achieved a maximum accuracy of 86.22% [16]. Hira Hameed et al. reported a contactless British sign language detection system and classified five emotions with spatiotemporal features acquired employing a radar system. They used deep-learning models such as InceptionV3, VGG16, and VGG19 to achieve a maximum accuracy of 93.33% [17]. In [18], the authors presented a contextual cross-modal transformer module for the fusion of textual and audio modalities operated on IEMOCAP and MELD datasets to achieve a maximum accuracy of 84.27%. In [19], the authors illustrated a speech recognition technique on frequency domain features of an Arabic dataset using SVM, KNN, and MLP techniques to achieve a maximum recognition accuracy of 77.14%. In [20], the authors proposed a fusion model both at the feature level (with an LSTM network) and decision level for happy emotion and achieved a maximum accuracy of 95.97%. The non-physiological signals are maskable by the subject easily. Facial expressions can be controlled, as well as speech tones can be modulated intentionally.

### 2.2. Physiological Signal-Based Emotion Classifiers

Researchers have also explored the physiological method of emotion detection. Mainly, EEG and GSR signals have been used to develop classifier models. Even then, unimodal ECG-based human recognition systems offering high accuracy using contactless acquisition of the ECG signal are still not much explored. In contactless systems, ECG data acquisition with minimum artefacts has always been a challenge to researchers. This work proposes an ECG belt system for contactless acquisition capable of acquiring ECG data with high precision and accuracy. The functional block diagram of the ECG data acquisition system is shown in Figure 2.

M. R. Islam et al. conducted an extensive review of EEG-based emotion recognition techniques in two categories, deep-learning- and shallow-learning-based models. A very detailed list of features used by researchers for the development of emotion classification models was reported. The paper analyzed the relevance of features, classifier models, and publicly available datasets. The authors minutely identified the advantages, as well as the issues, of each technique reported in the domain and suggested possible methods to overcome them [21]. E. P. Torres et al. illustrated RF and deep-learning algorithms to classify emotional states in stock trading behavior using features (five frequency bands, DE, DASM, and RASM) of an EEG signal. The relevance of each feature was identified by a chi-square test, and a maximum accuracy of 83.18% was achieved [22]. T. Song et al. developed a multimodal physiological signal database which includes EEG, ECG, GSR, and respiration signals. Some video clips were selected to induce emotions, and SVM and KNN were used to classify emotions. Moreover, they proposed a novel A-LSTM for more distinctive features and shared their database publicly. The Spearman correlation coefficient was used to identify negative and positive correlated emotions [23]. L. D. Sharma et al. used publicly available databases of ECG and EEG signals. Features were extracted using decomposition into reconstructed components using sliding mode spectral analysis techniques and machine-learning techniques for classification. Two publicly available databases, DREAMER and AMIGOS, were analyzed to achieve a maximum accuracy of 92.38% [24]. G. Li et al. used an EEG-signal-based SEED dataset and experimentally performed batch normalization. An LR classifier was implemented on PSD features of the EEG signals to improve the recognition accuracy of the system by up to 89.63% [25]. A. Goshvarpour et al. examined the effectiveness of the matching pursuit algorithm in emotion recognition problems. They acquired ECG and GSR data of 16 students (a smaller number of subjects) by exposing them to emotional music clips and developed an accurate emotion recognition system based on machine-learning classification tools (such as PCA-KNN) and discriminant analysis. They achieved a 100% accuracy rate and concluded that ECG is a more effective parameter to classify emotions in comparison with GSR [26]. The sample size taken was small. Huanpu Yin et al. presented a contactless IoT user identification and emotion recognition technique. A multi-scale neural network with a mmWave radar system was designed for accurate and robust sensing and achieved a maximum accuracy of 87.68% [27]. Alex Sepulveda et al. established the use of ECG signal features extracted from the AMIGOS database using wavelet transform and classified emotions using KNN, SVM, ensemble, etc., to achieve a maximum accuracy of 89.40% [28]. Muhammad Anas Hasnul et al. reviewed emotion recognition systems based on ECG signals and established the emotional aspect of various heart nodes. They highlighted systems with an accuracy of more than 90% and also validated the publicly available databases [29]. In [30], the authors included a comment on the relationship between emotional state and personality traits in which EEG, ECG, and GSR with facial features were utilized to establish a non-linear correlation. In [31], the authors proposed a deep fusion multimodal model to enhance the accuracy of class separation for emotion recognition on DEAP and DECAF datasets and achieved a maximum accuracy of 70%. An emotion recognition system with multiple modalities such as facial expression, GSR, and EEG using LUMED-2 and DEAP datasets to classify seven emotions with a maximum accuracy of 81.2% has been reported [32]. In [30], the authors reported a hybrid sensor fusion approach to develop a user-independent emotion recognition system. WMD-DTW (a weighted multi-dimensional DTW) and KNN were used on E4 and MAHNOB datasets to achieve a maximum accuracy of 94% [33]. A real-time IoT and LSTM-based emotion recognition system utilizing physiological signals was reported to classify emotions by up to an F-score of 95% achieved using deep-learning techniques [34]. Many of the above have either preferred using publicly available data or the subjects they have investigated have been too few. The classification accuracies achieved in the reported work have room for improvement. The subjects investigated in [26] are too few to claim 100% accuracy in which the classifier generalization may be questionable.

In this work, our purpose was to investigate a greater number of subjects from a wide range of age groups to develop a highly accurate ECG-based emotion classifier system. The classification accuracies can be enhanced using SVM and RF as achieved in this proposed work, and the comparative study is tabulated in Table 1. This article presents an economical, simple, and highly accurate contactless ECG-based human emotion recognition system. The ECG waveform corresponding to an induced emotion was acquired using an ECG belt system [35] which was received by the PC via a serial port using a Bluetooth module.

The acquired signal was processed by the signal processing block using MATLAB 2020b software. A total of ten statistical features of the ECG signal were extracted for each subject for all seven emotions in an induced emotional state while watching videos containing specific emotional content. Processing included sampling, normalization, denoising, and the removal of outliers to obtain processed ECG data in .csv file spreadsheets. Finally, the prepared datasets were used to extract features using feature extraction algorithms developed using MATLAB codes. The feature datasets were used for training and testing for the development of emotion classifiers using machine-learning tools. The models were trained and cross-validated to classify different emotions. The emotion classifier models were designed to classify seven types of emotions: Anger, Disgust, Fear, Happy, Neutral, Sad, and Surprise.

These models offer higher classification accuracy in comparison to the unimodal physiological-signal-based emotion recognition systems reported earlier. The contactless ECG belt acquisition system offers an accurate, compact, and contactless ECG data acquisition which does not require on-body AgCl gel-based ECG electrodes. So, the proposed system differs from other contactless systems such as radar-based systems which utilize the electromechanical movement of the chest and are usually complex and bulky. The system response is accurate and has the potential to be developed as an end-use product such as a lie detector. The key contributions of the proposed work are as follows:A hardware system for an accurate ECG acquisition from above the cloth has been designed.Ten statistical features of the ECG data have been extracted using developed feature extraction algorithms.Experiments have been conducted on a total of 45 subjects and seven different categories of emotions using emotion classifier models (such as SVM and RF) developed using MATLAB have been classified.Developed models are capable of classifying human emotions precisely with reasonably high classification accuracies for a contactless unimodal ECG-based emotion recognition system.

## 3. Methodology

The proposed system includes an efficient wireless ECG data acquisition belt system utilizing capacitive sensing technology (offers an accuracy of almost 96%), signal processing block (filtration, normalization, noise removal, and removal of outliers) and feature extraction algorithm (F1–F10), emotion classification using classifier models (SVM and RF classifiers), feature selection, and validation. A detailed explanation of each block is included in the later part of the section. The experiment involved human subjects and hence was conducted very cautiously; some of the experimental data took many iterations before being considered in the final dataset. The whole process flow chart is shown in Figure 3. It involves the ECG data acquisition of subjects corresponding to a specific emotion. The acquired data is pre-processed (normalization, denoising, and removal of outliers) using the signal processing module of MATLAB R2020b. The statistical features (F1–F10) of the ECG data are calculated using a feature extraction algorithm developed and running as a live script. The feature dataset acquired for each set of subjects is used to train models using classifier learners. The emotion classifiers (SVM and RF) are developed as well as hyperparameters are tuned using optimizable forms.

The major implementation challenges are listed below:A contactless method of accurate ECG acquisition is in itself a challenging task. The designed hardware acquires ECG from above the cloth of the subject. The noise minimization was achieved using filters designed using low-noise op-amps. The acquired signal was processed in the signal processing module.The stable connection of the Bluetooth module (HC-05) with the PC system was a challenge while acquiring the data. Experiments were required to be repeated in some cases.One of the features (heart rate) extracted for the ECG signal for the corresponding emotion was verified with a Mi Band 5 simultaneously. In some cases, the experiment was repeated.The intensity of any emotion also depends on the diurnal baseline of an individual mood; the baseline was modified as a neutral case for each individual. It can be referred to as calibration before experimentation.

The experimental steps followed in this study are as follows:The volunteers who participated in this experimental study were explained the methodology, and the required consent was taken. The volunteers were healthy subjects with no cardiac history.The volunteers were categorized into three different groups based on the age group to which they belonged. The three different age groups selected were 21–30 years, 31–40 years, and 41–50 years, with a male to female of 3:2.The audiovisual stimuli were already prepared in the form of video clips of 60 s length for each of the seven different emotions inclusive of pre- and post-10 s of static image to obtain a baseline. The selection of the audiovisual stimuli was made with the help of the previous literature [39,40,41,42].A total of 45 volunteers (15 from each of the selected age groups) were shown the audiovisual stimuli in random order but while keeping an average time gap of 10 min between two video stimuli or until the subject returned to their baseline level.While watching the video to elicit a specific emotion, the ECG of the subject from the ECG belt system was recorded and sent in parallel wirelessly to MATLAB using the serial port of the PC.An ECG feature extraction algorithm was developed; a live script ran on MATLAB in which the acquired ECG signal was firstly processed for noise removal and normalization, and then the statistical features were extracted.The subjects were also asked to write down the emotions they felt while watching the audiovisual stimuli; only those cases are considered in which the category of the clip and the emotion felt and mentioned by the subject matched.One of the primary features, heart rate, calculated from the feature extraction algorithm was verified with the heart rate acquired using the PPG sensor of a Mi Smart wrist Band 5. The experiment was repeated for the case in which the heart rate using the algorithm and smart band varied by more than 3–5 bpm.The ECG data for all seven emotions for the selected age groups and the corresponding feature datasets in the form of three different sets (Set A, Set B, and Set C) were prepared as final feature datasets.The datasets were used individually to train classifier models using SVM and RF classifiers with a 10-fold cross-validation technique with PCA enabled.Two principal components were selected for each set of data to avoid overfitting problems. The significant features were selected using the wrapped method.The hyperparameters were tuned using the Bayesian optimization technique using optimizable classifier learners for both of the designed models to recommend the best hyperparameters.The classification accuracy obtained was validated and compared with available techniques.

### 3.1. Data Acquisition

The ECG data was acquired using a non-contact capacitive ECG belt system. It is a 2-lead ECG system with a reference electrode attached to the wrist. The belt was wrapped around the waist above the cloth only. It can acquire the ECG of a person above a cloth of thickness up to 100 microns. The subjects under investigation were exposed to video clips of 40 s length for 7 different emotions: Anger, Disgust, Fear, Happy, Neutral, Sad and Surprise. The data acquisition system hardware included a signal processing block containing an instrumentation amplifier implemented using INA128P for signal amplification and filters using low noise op-amp LM324N to remove noises such as motion artefacts, etc. The overall accuracy of the 2-lead ECG belt system was compared with a 12-lead Holter monitor, which is quite high (up to 96%), and it was found very suitable for non-contact ECG acquisition [35]. To minimize the noise introduced due to the AC power line, it was ensured that the devices were not directly connected to the power line but made to run on a DC supply. Moreover, the signal processing block contained a notch filter of 50 Hz to suppress power line frequency. To avoid other common noise introduced by the lead electrodes (AT3 coaxial probe, Schiller), muscle movements, and other motion artefacts, an HPF and an LPF were designed to select the signal frequency of 0.49 Hz to 100 Hz using low noise op-amps (LM324 AD, Texas Instruments). The common noise introduced due to muscles, etc. was handled by a high CMRR (125 dB) offered by an instrumentation amplifier (INA128P, Texas Instruments) [32].

The signal processing block was connected to an Arduino controller which controls the sensing application and data acquisition. The acquired data were transferred using a Bluetooth module to the PC for further processing and feature extraction. The block diagram of the data acquisition system is shown in Figure 2. The acquired ECG signal obtained for a subject in a neutral state is shown in Figure 4.

### 3.2. Preparation of ECG Datasets

For each set of data, the ECG signal corresponding to induced emotions for 15 subjects was recorded and analyzed. Each person was shown all 7 video clips containing 7 different emotions, and the normal state was ensured by keeping a gap of 10 min between the videos. Three sets of data, named Set A, Set B, and Set C, were prepared. Set A was acquired for the subjects within the age group of 21–30 years, Set B for the 31–40 years age group, and Set C for the 41–60 years age group. The male-to-female ratio was 3:2. The subjects volunteered and were explained the experimental procedure, and the required consent was taken. The demographic details of the subjects are included in Table 2.

### 3.3. ECG Data Processing

The ECG data were obtained for all 15 subjects in .csv format with a gain of 100 and a sampling frequency of 256 Hz. The data were recorded for 60 s including pre- and post-ten seconds of normal state value for the baseline. To ensure the base value, a static image of natural scenery was shown. The subject was exposed to video clips of 40 s length, hence the ECG data for that period were taken out for feature extraction and emotion classification. The data were processed to remove any noise using the denoising application of the signal processing tool in MATLAB R2020b; one of the cases is shown in Figure 5. The denoised signal was then filtered using the high-pass and low-pass filters simultaneously to clip off those frequency components which contained noise. The amplitude level of the signal was normalized, and outliers were removed. Some samples of the processed ECG data of one subject for a neutral state are shown in Figure 6.

### 3.4. Feature Extraction

The main idea behind the development of a unimodal ECG-based emotion recognition system is the potential of the physiological signals to have information that can be used to classify the emotional state of a person as is claimed in much of the reported literature. The ECG signal is a quasi-periodic bioelectrical signal of low amplitude and frequency. The minor variation in the amplitude of the signal will majorly constitute the emotional contents. The contactless method was utilized in this work, and the ECG signal was processed properly to minimize the impacts of motion artefacts and any other system noises. Theoretically, the statistical features selected represent those minor variations in the ECG signal. Experimentally, the partial dependence plots were plotted to interpret the models’ classification dependency on a particular feature. It was found that each model had statistical ECG features with an average contribution of 8%, except for heart rate which contributed 15% on average. The minor fluctuations in the ECG signal can be expressed in terms of the statistical features of the ECG data to train emotion classifier models.

For 40 s of ECG data sampled at 256 Hz, for one emotion and a single subject, there was a total of 10,240 data points. The parameters were extracted for each sampling interval, hence for one subject and one emotion; the feature set was of a 40 × 10 array. For 15 subjects, each feature dataset had a size of 4200 × 10, corresponding to the ECG dataset of 1,075,200 data points as tabulated in Table 3.

A total of 10 statistical time-domain features (F1–F10) of ECG signals were calculated. The features (mean, standard deviation, variance, minimum value, maximum value, first-degree difference, kurtosis, skewness, heart rate, and RRmean) were extracted from .csv data files of the recorded ECG signals for the various emotional states of the subjects. The parameters and their mathematical expressions are tabulated in Table 4. The parameters were calculated for each sampling time interval.

### 3.5. Classifier Learners

The extracted features for all selected emotional states were prepared with the corresponding emotional response allotted to them. The features were calculated using ECG data used to classify the emotional state of a person in this study. The datasets were prepared using features as input and emotional responses (seven categories) as output. The dataset was analyzed using various classifiers available in MATLAB R2020b. The data for each set were disjoint for the training set, and no data from other sets were used to train models except in one case when the combined data of all three sets were used to train the models separately.

#### 3.5.1. SVM

SVM is one of the most efficient supervised classifier learners. It divides the data into support vectors and keeps one vector of the data at a time for validation, trains the rest of the vectors, and this process repeats till the learner finds the most suitable decision boundary (hyperplane) to classify the data precisely and accurately. In this study, an SVM classifier with a quadratic kernel function was used to develop the emotion recognition models. To tune the hyperparameters, Bayesian optimization techniques with multiclass methods one-vs-one and one-vs-all were used. Different kernel functions in SVM optimizable (kernel functions: Gaussian, linear, quadratic, and cubic; the number of iterations: 30) were used in this research work to recommend the best hyperparameter. Moreover, to avoid overfitting the problem, principal component analysis (PCA) with a 95% variance level was enabled, and it selected 2 principal components.

#### 3.5.2. Random Forest

Random forest was one of the robust techniques used to classify the data more accurately and precisely. Ensemble bagged tree and, to tune hyperparameters, the ensemble optimizable (kernel functions: bag, AdaBoost, RUSBoost; optimizer: Bayesian optimization; the number of iterations: 30, PCA enabled) were the classifier learners used from this category. The ensemble bagged tree divides the data into subsets randomly with replacement to train and reduces the variance of the decision tree. It bags various models in one to enhance classification accuracy. Each subset is used to train the collection of decision trees and provide benefits for all decision trees. Hence, it is an ensemble of many decision trees operated on randomly divided subsets and bags the average response from all the decision trees to arrive at the most efficient classification models. Random forest, an extension of ensemble bagging, uses a random selection of features also to grow trees.

Classifiers used in this research work are SVM, RF, and their optimizable forms. SVM quadratic and ensemble bagged tree were the two machine-learning classifiers used for training and testing. Both of the techniques support classification as well as regression. In SVM, each parameter was correlated with other parameters individually and collectively to assess the correlation of features with the emotional response of the subject. It selected the best decision boundary, also called a hyperplane. RF contains multiple decision trees which combine to arrive at a single answer. A 10-fold cross-validation technique was used for each case. Optimization in both cases utilized 30 iterations to recommend hyperparameters and selected the best one.

## 4. Results and Discussion

The three ECG feature datasets were prepared out of three different raw ECG data (acquired using an ECG belt system hardware) for three different age groups. They have been used to develop models capable of classifying human emotions quite precisely and accurately. The experimental setup with a subject under investigation (permission granted to publish the photograph of the subject) is shown in Figure 7. The value of heart rate obtained in each iteration was verified using a Mi Smart Band 5 (Manufacturer: Xiaomi Inc., Beijing, China) with a PPG heart rate sensor and capacitive wear monitoring sensor. The emotion classifier models were trained, and the proposed models were validated using a ten-fold cross-validation technique with enabled PCA using MATLAB. Initially, the ECG feature datasets acquired for all three different age groups (Set A, Set B, and Set C) were used individually to develop classifier models. The classifiers used are SVM (kernel function: quadratic; kernel scale: automatic; multiclass method: one-vs-one) and random forest (ensemble method: bagged, learner type: decision tree method). The developed models were given the training and testing data from the completely disjoint sets. To analyze the generalizability of the models, the combined feature dataset (total of 45 subjects) was also used to train different sets of models using the selected classifier learners. The Bayesian optimization technique with expected the improvement per second plus acquisition function was used to tune the hyperparameters of the trained models in 30 iterations using optimizable versions of the classifiers. The minimum classification error point and recommended best hyperparameter for each of the datasets and combined datasets were obtained.

To analyze the performance metrics of the developed classifier models, the true class, the predicted class, the true positive rate (TPR) [40] and the false negative rate (FNR) plots corresponding to each emotion for all models trained using individual datasets and the combined datasets in the form of confusion matrices were acquired for both the classifiers. The matrices corresponding to each set and combined set with the SVM quadratic classifier are shown in Figure 8a–d, respectively. The confusion matrices for Set B of the ECG data have been included in the form of a total number of observations instead of percentages. It can be observed that the classifier is capable of predicting a neutral emotion state for each set with more than 99% accuracy. Set A has a maximum FNR of 9% for Sad misclassified as Fear. For the proposed model, the SVM quadratic classifier can classify emotions using ECG signal features with an average accuracy of 96.6%. The models trained using the SVM quadratic classifier model with Set B data offer a maximum classification accuracy of 86.6%. The SVM model trained with Set C offers a maximum FNR of 3% for Sad misclassified as Fear. The SVM model trained and validated with a combined ECG dataset offers a maximum classification accuracy of 94.8%. The experiments were conducted in a laboratory-based environment to develop this prototype system, and it was feasible to go for many iterations in some cases to acquire the required data. Moreover, the classifier models need to be assessed in a variety of real-world settings to ensure their generalizability because there may be various other factors that influence emotional state, hence the baseline. Although the ECG-based emotion classifier models developed in this work have been tested with some of the publicly available ECG datasets, the performance of the physiological-signal-based (ECG: involuntary in generation) classifier system was found to be promising and accurate.

The hyperparameters were tuned using the Bayesian optimization technique for the multiclass method (one-vs-one and one-vs-all), and four models of SVM optimizable with individual datasets and combined datasets were trained and developed. The confusion matrices and classification error plot with 30 iterations of optimization using the Bayesian method are shown in Figure 9A–D. The optimizable models utilize all the advantages of the available kernel functions to come up with recommendations for the best hyperparameter. Most of the models offer quadratic as the best kernel function for the minimum classification error. It can be noted that the overall accuracy of the models increased or was equal to the quadratic SVM model. They were trained to obtain the most suitable kernel function for emotion classification using unimodal ECG datasets.

The confusion matrices for the random-forest-based classifier named ensemble bagged tree classifier developed for each dataset and the combined are shown in Figure 10a–d. The TPR and FNR plots in their percentage of accuracies corresponding to the predicted class with true class are clearly shown in Figure 10. It can be observed that the ensemble bagged tree classifier is capable of predicting a neutral emotion state for each set with 100% accuracy. Corresponding to Set A data, the trained model has a maximum TPR of 100% for three emotions and a minimum accuracy of 94.6% for Sad, with a maximum FNR of 5.4% for Sad misclassified as Fear. The model designed with the Set B dataset has a maximum FNR of 0.8% and an overall classification accuracy of 99.1%. With the Set C dataset, the model offers a minimum classification accuracy of 99%. For the model designed using the combined set of data for testing and training data in the cross-validation technique, the overall classification accuracy is 99.6%. This proposed model, the ensemble bagged tree classifier, is capable of classifying human emotions using ECG signal features with a minimum average accuracy of 96%, which is higher than the SVM quadratic. The model-wise and set-wise average classification accuracies obtained are tabulated in Table 5. The models offering minimum and maximum accuracies for each feature dataset are marked in blue and orange bold colors, respectively. Many models using the ensemble bagged tree classification algorithm are capable of classifying at least one emotion completely.

The ensemble optimizable models were also trained and validated to obtain the most suitable kernel function offering minimum classification error and recommended the best hyperparameter. The confusion matrices and classification error plots with 30 iterations of optimization using the Bayesian method are shown in Figure 11A–D. Most of the models offer the bagged tree method as the best for the minimum classification error.

The designed system with reported classification accuracy was obtained experimentally for the emotion recognition of 45 healthy subjects of three different age groups. Further experimental analysis is needed to claim the generalization of the developed models for patients with cardiac issues. This work will be extended further in the future to analyze the emotions of patients with CVDs.

The ROC and AUC curves for all the models were analyzed, and they were found to be quite close to the ideal case. The ROC–AUC curve for the SVM model with the quadratic kernel function trained and tested using the Set A feature dataset (youngest of the age groups taken) is included in Figure 12. The plots show one of the emotions as a positive category and the rest as a false positive rate. The AUC is also indicated for one of the models trained.

The training time and prediction speed for all the training models used in this research work are tabulated in Table 6. The minimum classification error for both classifiers in optimizable learning models is also mentioned. The greater the prediction speed in observations per second, the less testing time the model needs to classify emotions for the new set of ECG feature datasets.

The overall performance of the random forest ensemble bagged tree classifier was found to be even better than the highly efficient SVM quadratic in the case of the models being designed using the same dataset, and the same goes for the optimizable classifiers of both classes. The designed system is a simple, economical, unimodal ECG-based emotion classifier system. A comparison bar graph showing the classification accuracies of all the developed emotion classifiers is shown in Figure 13. The results obtained are very promising to be realizable as an end-use product in areas such as the diagnosis of mental illness, the preferences of customers and buyers, analysis of the behavior patterns of investors, home automation, and lie detector systems.

## 5. Conclusions

The ECG signal containing the emotional contents of 15 subjects from each of the three different age groups was acquired using a non-contact capacitive ECG belt system. The signal was processed, and a total of ten features were extracted for each sample interval of the final sampled ECG signal datasets. The emotion classifiers were developed using SVM- and RF-based machine-learning techniques. The data were analyzed set-wise as well as subject-wise. The overall average classification accuracy achieved is very promising and sufficiently high. The proposed system and selected features are very suitable to detect the emotional state of a person using an ECG signal. The system is simple, very fast, and highly reliable. The response of the system was compared with available systems reported in the literature as tabulated in Table 1. To the best of the authors’ knowledge, the proposed system offers the highest accuracy by employing a non-contact ECG system using a unimodal signal with statistical features. The system is a state-of-the-art and highly efficient emotion classification system which has immense potential to contribute to the development of human–computer interaction systems.

## Figures and Tables

**Figure 1 diagnostics-13-02097-f001:**
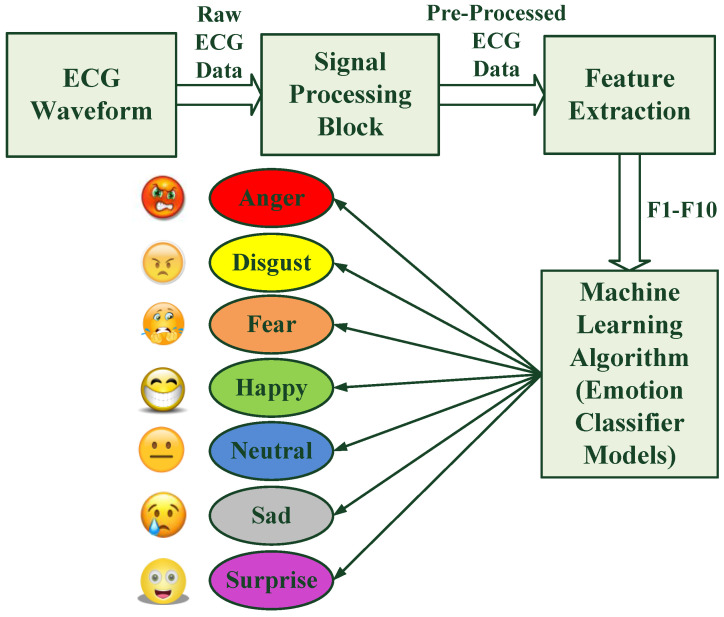
Functional block diagram of the proposed human emotion recognition system.

**Figure 2 diagnostics-13-02097-f002:**
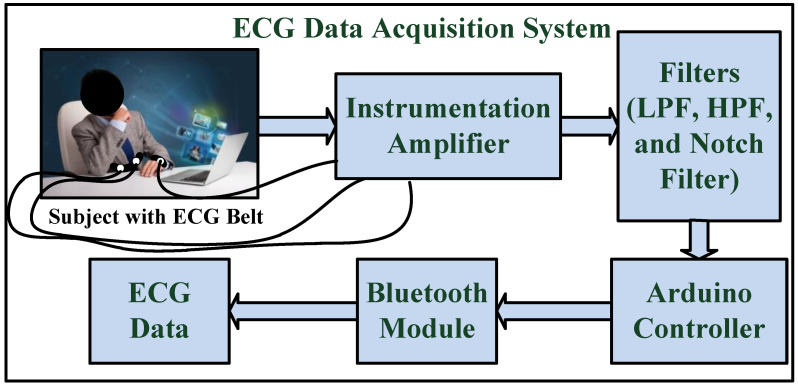
Block diagram of the ECG data acquisition system.

**Figure 3 diagnostics-13-02097-f003:**
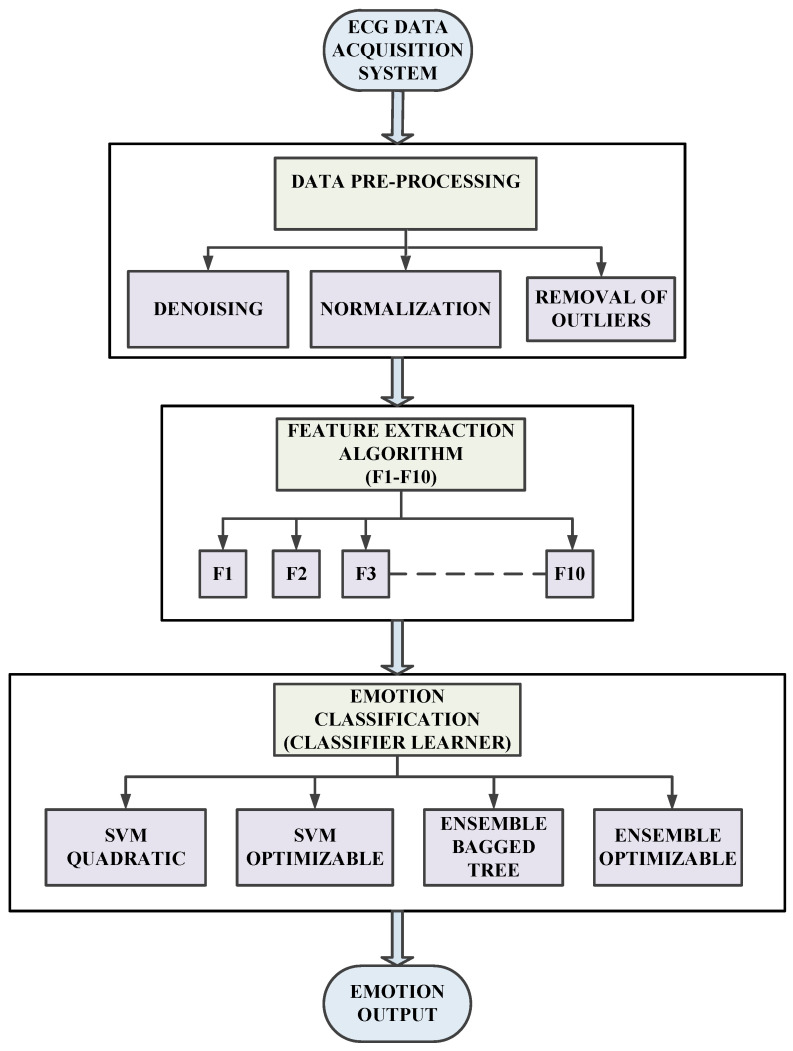
Process flow chart of the emotion recognition model.

**Figure 4 diagnostics-13-02097-f004:**
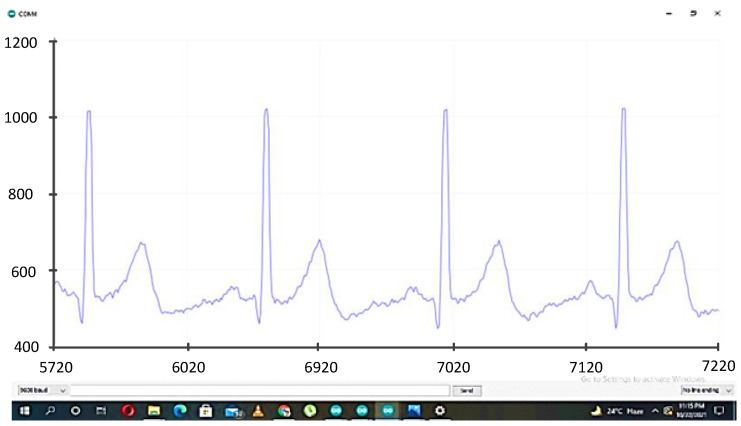
ECG signal on serial plotter of Arduino IDE.

**Figure 5 diagnostics-13-02097-f005:**
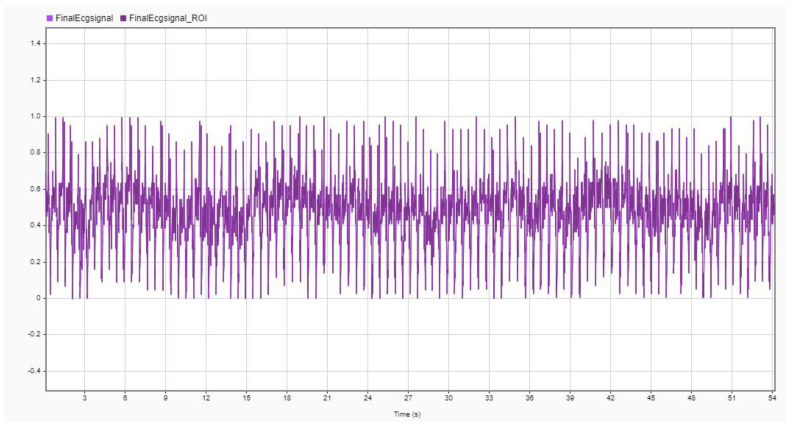
A normalized ECG signal in time-domain.

**Figure 6 diagnostics-13-02097-f006:**
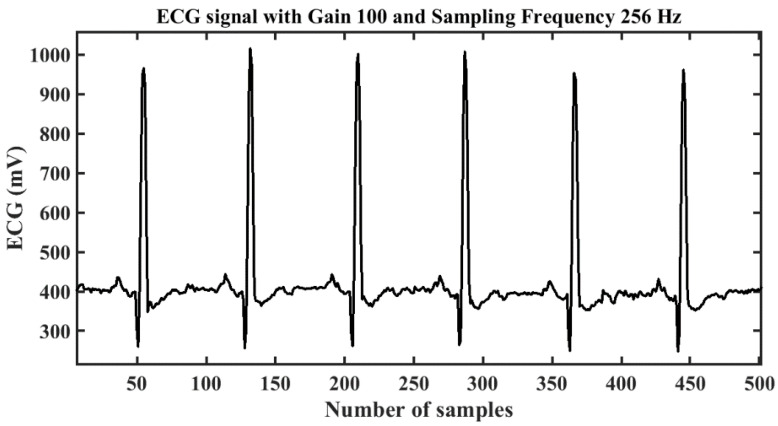
The pre-processed sampled ECG signal.

**Figure 7 diagnostics-13-02097-f007:**
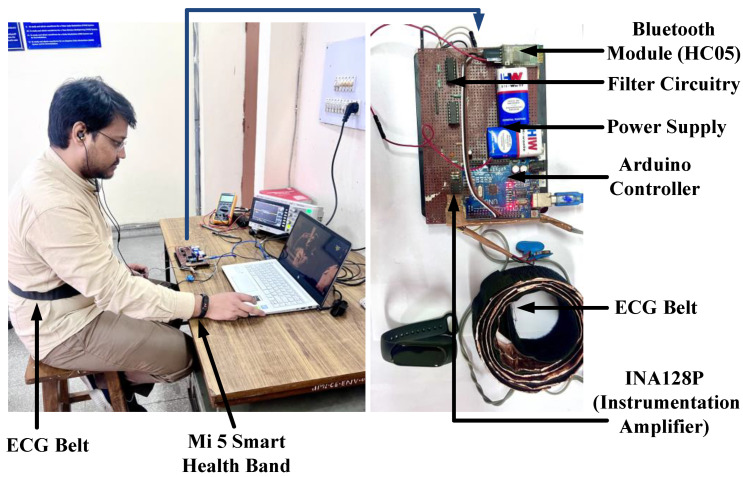
Snapshot of the experimental setup.

**Figure 8 diagnostics-13-02097-f008:**
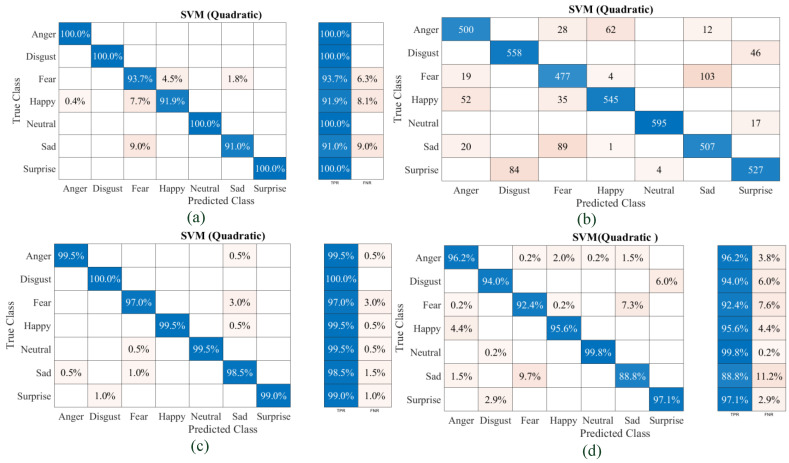
Confusion matrices of the SVM quadratic classifier: (**a**) Set A feature dataset; (**b**) Set B feature dataset; (**c**) Set C feature dataset; (**d**) combined feature dataset.

**Figure 9 diagnostics-13-02097-f009:**
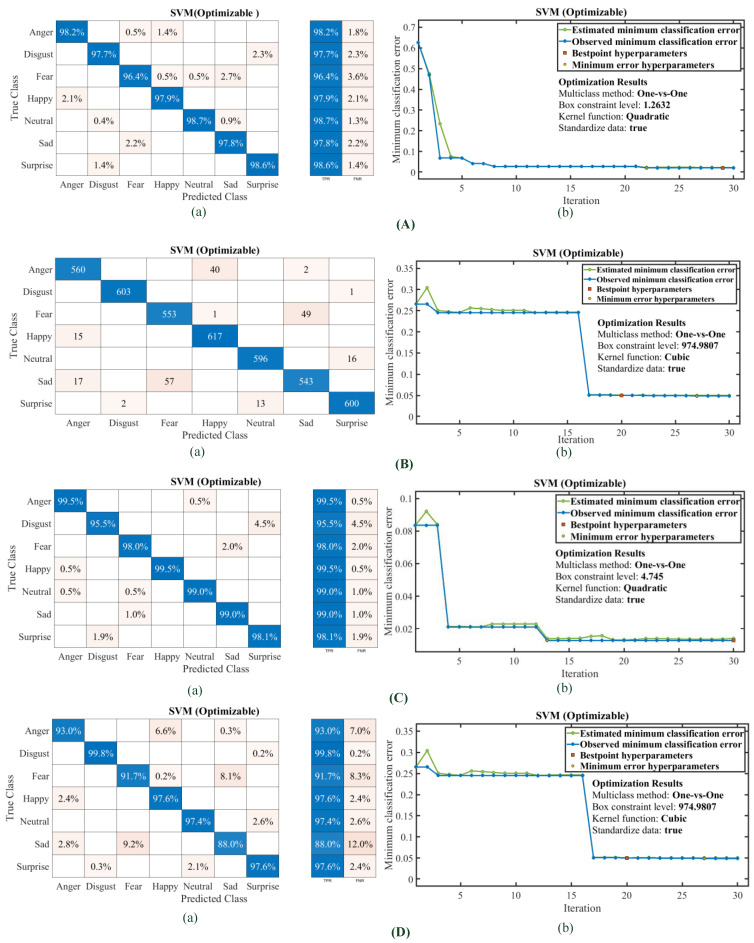
Confusion Matrices of SVM Optimizable and minimum classification error plot (**A**). Set A Feature dataset: (a) confusion matrix (b) minimum classification error plot (**B**). Set B Feature dataset: (a) confusion matrix (b) minimum classification error plot (**C**). Set C Feature dataset: (a) confusion matrix (b) minimum classification error plot (**D**). Combined Feature dataset: (a) confusion matrix (b) minimum classification error plot.

**Figure 10 diagnostics-13-02097-f010:**
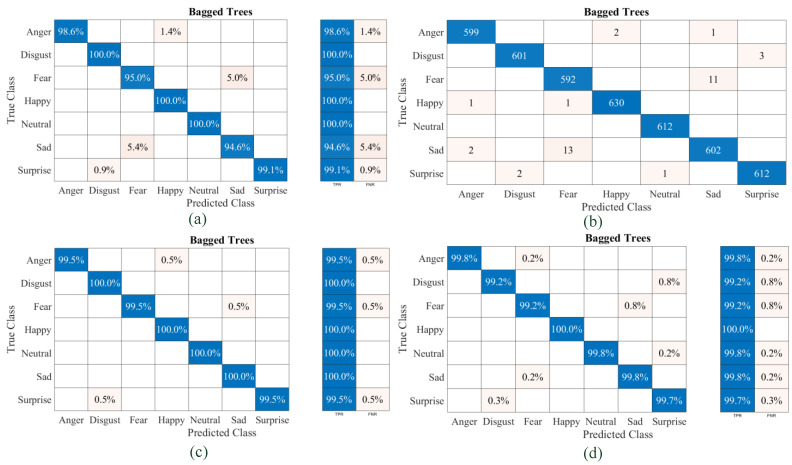
Confusion matrices of the ensemble bagged tree classifier: (**a**) Set A feature dataset; (**b**) Set B feature dataset; (**c**) Set C feature dataset; (**d**) combined feature dataset.

**Figure 11 diagnostics-13-02097-f011:**
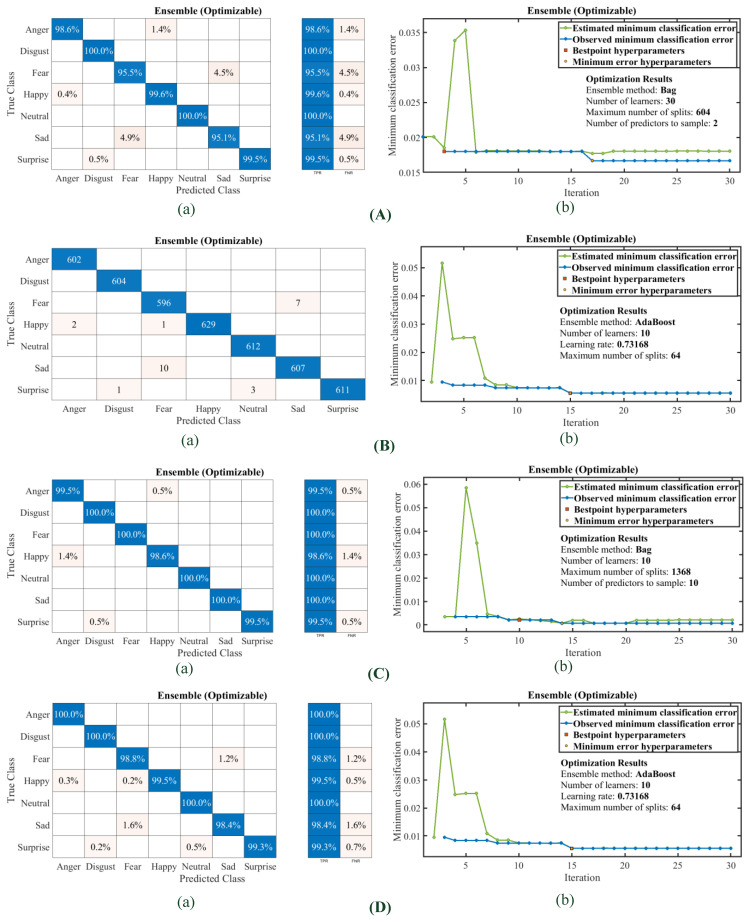
Confusion Matrices of Ensemble Optimizable and minimum classification error plot (**A**). Set A Feature dataset: (a) confusion matrix (b) minimum classification error plot (**B**). Set B Feature dataset: (a) confusion matrix (b) minimum classification error plot (**C**). Set C Feature dataset: (a) confusion matrix (b) minimum classification error plot (**D**). Combined Feature dataset: (a) confusion matrix (b) minimum classification error plot.

**Figure 12 diagnostics-13-02097-f012:**
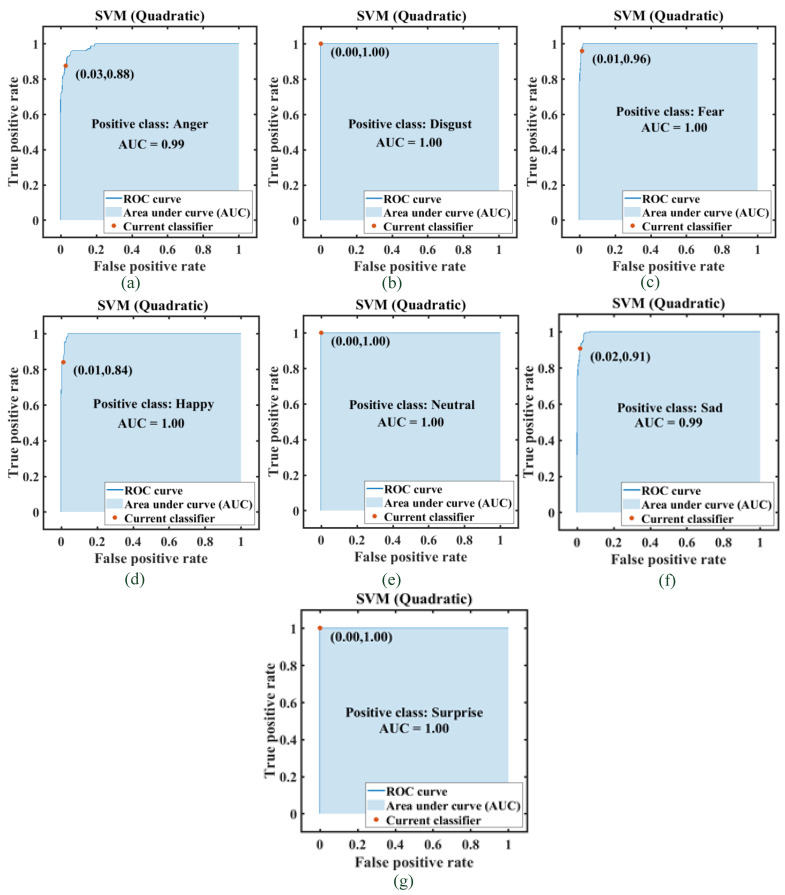
AUC- ROC plot for all seven categories of emotions in case of SVM Quadratic model (Set-A). (**a**) positive class: Anger (**b**) positive class: Disgust (**c**) positive class: Fear (**d**) positive class: Happy (**e**) positive class: Neutral (**f**) positive class: Sad (**g**) positive class: Surprise.

**Figure 13 diagnostics-13-02097-f013:**
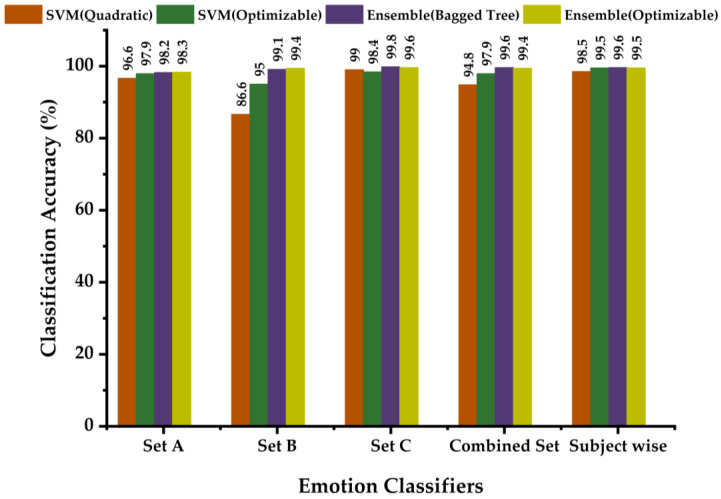
Comparison bar graph of classification accuracies for emotion classifier models.

**Table 1 diagnostics-13-02097-t001:** Comparative Study.

Reference	Technique	No. of Features	No. of Emotions	Max Accuracy	Contribution	Limitations	Year
[13]	RF	Speech datasets	8	89.60%	Emotion recognition from speech signals using WPT cochlear filter bank and random forest classifier	A non-physiological signal can be manipulated by the subject	2021
[15]	AdaBoost-KNN	Facial expression dataset	7	94.28%	AFS-AdaBoost-KNN-DO; a dynamic facial emotion recognition	Emotions can be masked in a facial expression-based technique	2021
[25]	KNN, SVM, RF, LR, BN, MLP, CNN, etc.	2 (PSD and DE)	3 (positive, neutral, and negative)	89.63% when using the LR classifier	Real-time emotion recognition system using an EEG signal	Averaging in preprocessing resulted in a loss of emotional information	2022
[26]	LDA, Neural Network	5	4	94.75%	An accurate emotion recognition using ECG and GSR	Less number of features	2017
[36]	SGCN (CNN)	4	4	96.72%	A deep hybrid network called ST-GCLSTM for EEG-based emotion recognition	Fewer features	2022
[37]	Parameterized photograms and machine learning	Facial expression dataset	7	99.80%	High accuracy using weak classifiers on publicly available datasets	Facial expression can be controlled by the subject	2021
[38]	SVW, CART, and KNN	8	4	97%, (Angry)	ECG signal correlation features in emotion recognition	Accuracies are less for all emotions except ‘angry’	2020
Proposed work	SVM and RF	10	7	98%	ECG-based emotion classifier model, highly accurate, high prediction speed	Signals such as EEG, EMG, BR, and skin conductance can be explored as a unimodal feature	-

**Table 2 diagnostics-13-02097-t002:** Demographic details of the subjects.

Dataset	Number of Subjects	Male:Female Ratio	Average Age (yrs.)
A	15	3:2	23
B	15	3:2	36
C	15	3:2	44
Combined	45	3:2	34

**Table 3 diagnostics-13-02097-t003:** Size of ECG datasets.

Dataset Type	Typical Size of the Datasets			
	ECG Data (per Emotion per Subject)	ECG Data Points (per Subject)	ECG Data Points (for All Subjects)	ECG Feature Dataset
Set A	10,240	71,680	1,075,200	4200 × 10
Set B				
Set C				
Combined Set			3,225,600	12,600 × 10

**Table 4 diagnostics-13-02097-t004:** Selected statistical parameters for feature selection and their mathematical expression.

S. No.	ECG Feature		Mathematical Expression N = Number of Samples, Sampling Frequency fs = 256 Hz Signal Duration, T = 40 s
	Name	Parameter	
1.	F1	Mean	$\frac{1}{N}\overset{N}{\underset{n=1}{\sum }}x_{n}$
2.	F2	Standard Deviation	$\sqrt{\frac{1}{N}\overset{N}{\underset{n=1}{\sum }}{\left(x_{n}-\mu_{x}\right)}^{2}}$
3.	F3	Variance	$\frac{1}{N}\overset{N}{\underset{n=1}{\sum }}{\left(x_{n}-\mu_{x}\right)}^{2}$
4.	F4	Minimum Value	min(X)
5.	F5	Maximum Value	max(X)
6.	F6	Skewness	$\frac{\sqrt{N\left(N-1\right)}}{N-2}\frac{\sum_{n=1}^{N}{\left(x_{n}-\mu_{x}\right)}^{3}∕N}{s^{3}}$
7.	F7	Kurtosis	$\frac{\sum_{n=1}^{N}{\left(x_{n}-\mu_{x}\right)}^{4}∕N}{s^{4}}$
8.	F8	First-Degree Difference	$\frac{1}{N-1}\overset{N-1}{\underset{n=1}{\sum }}\left|x_{n+1}-x_{n}\right|$
9.	F9	Heart Rate (BPM)	$\frac{Beat\;count}{Signal\;duration\;in\;minutes}\;$
10.	F10	RRmean	$\frac{Signal\;duration\;in\;seconds}{Heart\;rate\;\left(BPM\right)}\;;$ $Duration\;in\;seconds=\frac{N}{f_{s}}\;$

**Table 5 diagnostics-13-02097-t005:** Average classification accuracy of the proposed models.

Dataset Type	Classifier	Model Type	Average Accuracy in Emotion Recognition (%)
Set A ECG Feature Data	Support Vector Machine (SVM)	** Quadratic **	** 96.6% **
		Optimizable	97.9%
	Random Forest (RF)	Ensemble Bagged Tree	98.2%
		** Ensemble Optimizable **	** 98.3% **
Set B ECG Feature Data	Support Vector Machine (SVM)	** Quadratic **	** 86.6% **
		Optimizable	95.0%
	Random Forest (RF)	Ensemble Bagged Tree	99.1%
		** Ensemble Optimizable **	** 99.4% **
Set C ECG Feature Data	Support Vector Machine (SVM)	Quadratic	99.0%
		** Optimizable **	** 98.4% **
	Random Forest (RF)	** Ensemble Bagged Tree **	** 99.8% **
		Ensemble Optimizable	99.6%
Set A, B, and C combined ECG Feature Data	Support Vector Machine (SVM)	** Quadratic **	** 94.8% **
		Optimizable	97.9%
	Random Forest (RF)	** Ensemble Bagged Tree **	** 99.6% **
		Ensemble Optimizable	99.4%
Subject wise ECG Feature Data	Support Vector Machine (SVM)	** Quadratic **	** 98.5% **
		Optimizable	99.5%
	Random Forest (RF)	** Ensemble Bagged Tree **	** 99.6% **
		Ensemble Optimizable	99.5%

**Table 6 diagnostics-13-02097-t006:** Performance parameters of the proposed models.

Dataset Type	Classifier Model Type		10-Fold Cross-Validation		
			Average Training Time (Seconds)	Prediction Speed (obs/s)	Minimum Classification Error
Set A ECG Feature Data	Support Vector Machine (SVM)	Quadratic	5.078	12,000	-
		Optimizable	113.56	20,000	0.03
	Random Forest (RF)	Ensemble Bagged Tree	3.4884	8900	-
		Ensemble Optimizable	111.58	21,000	0.00127
Set B ECG Feature Data	Support Vector Machine (SVM)	Quadratic	1.7669	16,000	-
		Optimizable	666.76	18,000	0.02
	Random Forest (RF)	Ensemble Bagged Tree	2.1453	10,000	-
		Ensemble Optimizable	119.88	1900	0.0001
Set C ECG Feature Data	Support Vector Machine (SVM)	Quadratic	1.8198	14,000	-
		Optimizable	492.72	20,000	0.015
	Random Forest (RF)	Ensemble Bagged Tree	2.2686	11,000	-
		Ensemble Optimizable	108.32	2100	0.0007
Set A, B, and C combined ECG Feature Data	Support Vector Machine (SVM)	Quadratic	8.5107	22,000	-
		Optimizable	2110.8	34,000	0.002
	Random Forest (RF)	Ensemble Bagged Tree	5.0359	16,000	-
		Ensemble Optimizable	181.74	7200	0.002
Subject-wise ECG Feature Data	Support Vector Machine (SVM)	Quadratic	1.4402	5900	-
		Optimizable	237.31	3700	0.0137
	Random Forest (RF)	Ensemble Bagged Tree	2.093	2900	-
		Ensemble Optimizable	98.879	5600	0.0125

## Data Availability

Not applicable.

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
