# Peer review of "Design and Development of a Non-Contact ECG-Based Human Emotion Recognition System Using SVM and RF Classifiers"

_diagnostics, 2023, doi:10.3390/diagnostics13122097_

Round 1
Reviewer 1 Report
1 “The key contributions of the proposed work are”, authors gave 9 points but that are too general. Novelty is confusing. The main contributions of the manuscript are not clear. The main contributions of the article must be very clear and would be better if summarize them into 3-4 points.
2 “Introduction”, section is too long, there should be subsection “related work”.
3 Introduction section is still weak. An introduction is an important road map for the rest of the paper that should be consist of an opening hook to catch the researcher's attention, relevant background study, and a concrete statement that presents main argument but your introduction lacks these fundamentals, especially relevant background studies. This related work is just listed out without comparing the relationship between this paper's model and them; only the method flow is introduced at the end; and the principle of the method is not explained. To make soundness of your study must include these latest related works. Authors also need to justify the importance of their article and cite all of them to make a critical discussion that makes a difference from others' work.
i Long Dialogue Emotion Detection Based on Commonsense Knowledge Graph Guidance. IEEE Transactions on Multimedia. doi: 10.1109/TMM.2023.3267295
ii SandplayAR: Evaluation of psychometric game for people with generalized anxiety disorder. The Arts in Psychotherapy, 80, 101934. doi: https://doi.org/10.1016/j.aip.2022.101934
iii Fast Visual Tracking With Siamese Oriented Region Proposal Network. IEEE signal processing letters, 29, 1437. doi: 10.1109/LSP.2022.3178656
4 Authors used SVM and RF classifiers but there are latest techniques of deep learning too.
5 “For each set of data, the ECG signal of 15 subjects has been investigated”, what does it mean?
6 “FIGURE 3” should be improve in quality and need to be given detailed description.
7 When writing phrases like “To analyze the performance metrics of the developed classifier models, the True Class, the Predicted Class, the True Positive Rate (TPR)”, it must cite related work in order to sustain the statement (https://doi.org/10.1155/2023/2345835).
8 Authors should mention the implementation challenges.
9 Mention the limitations and future works of the developed system elaborately.
Moderate editing of English language required
Reviewer 2 Report
See attached file.

Round 2
Reviewer 1 Report
The authors have answered my questions satisfactorily.
Minor editing of English language required.
Author Response
Dear Reviewer, We are highly thankful for your kind consideration of the response to the comments submitted in the first round. Thank you for your satisfaction with the modifications we made following your suggestions and recommendations.
Response to the comment:
Comment 1: Minor editing of English language required.
Response: Thank you for the recommendation. The manuscript has been checked thoroughly and the required updations have been made. The English grammar has also been checked using software tools.
We are highly grateful for your suggestions as they helped us improve the quality and value of the article. We acknowledge your efforts and time.
Thank you
Reviewer 2 Report
The Authors did a notable job, and the paper is much more valuable now. As far as I am concerned, the paper can be published now. However, I suggest the Authors to add in the paper the response they gave me to Point 5 of the previous review round.
Author Response
Dear Reviewer, We are highly thankful for your appreciation. Your suggestions and recommendations have really helped in revising the paper. We thank you for considering the responses submitted in the first round.
Response to the Round-2 Comments:
Comment: I suggest the Authors to add in the paper the response they gave me to Point 5 of the previous review round.
Response: Thank you for the suggestion. The response given in point 5 of the first round review is included in the manuscript as suggested. It is incorporated in the updated manuscript in lines 389-397, Section 3.4 (Feature Extraction) on Pages 11-12.
We are highly grateful for your suggestions as they helped us improve the quality and value of the article. We acknowledge your efforts and time.
Thank you